# Drug-Utilization, Healthcare Facilities Accesses and Costs of the First Generation of JAK Inhibitors in Rheumatoid Arthritis

**DOI:** 10.3390/ph16030465

**Published:** 2023-03-21

**Authors:** Irma Convertino, Valentina Lorenzoni, Rosa Gini, Giuseppe Turchetti, Elisabetta Fini, Sabrina Giometto, Claudia Bartolini, Olga Paoletti, Sara Ferraro, Emiliano Cappello, Giulia Valdiserra, Marco Bonaso, Corrado Blandizzi, Marco Tuccori, Ersilia Lucenteforte

**Affiliations:** 1Unit of Pharmacology and Pharmacovigilance, Department of Clinical and Experimental Medicine, University of Pisa, 56126 Pisa, Italy; 2Institute of Management, Scuola Superiore Sant’Anna, 56100 Pisa, Italy; 3Tuscan Regional Healthcare Agency, 50100 Florence, Italy; 4Medical Specialization School of Pharmacology, University of Pisa, 56126 Pisa, Italy; 5Unit of Medical Statistics, Department of Clinical and Experimental Medicine, University of Pisa, 56126 Pisa, Italy; 6Unit of Adverse Drug Reactions Monitoring, University Hospital of Pisa, 56126 Pisa, Italy

**Keywords:** JAK inhibitors, tofacitinib, baricitinib, rheumatoid arthritis, drug-utilization, cost, healthcare system, biologic, DMARD

## Abstract

This study is aimed at describing tofacitinib and baricitinib users by characterizing their prescription and healthcare histories, drug and healthcare utilization patterns, and direct costs from a healthcare system perspective. This retrospective cohort study was performed using Tuscan administrative healthcare databases, which selected two groups of Janus kinase inhibitors (JAKi) incident users (index date) from 1st January 2018 to 31 December 2019 and from 1 January 2018 to 30 June 2019. We included patients ≥18 years old, at least 10 years of data, and six months of follow-up. In the first analysis, we describe mean time, standard deviation (SD), from the first-ever disease-modifying antirheumatic drug (DMARD) to the JAKi, and costs of healthcare facilities and drugs in the 5 years preceding the index date. In the second analysis, we assessed Emergency Department (ED) accesses and hospitalizations for any causes, visits, and costs in the follow-up. In the first analysis, 363 incident JAKi users were included (mean age 61.5, SD 13.6; females 80.7%, baricitinib 78.5%, tofacitinib 21.5%). The time to the first JAKi was 7.2 years (SD 3.3). The mean costs from the fifth to the second year before JAKi increased from 4325 € (0; 24,265) to 5259 € (0; 41,630) per patient/year, driven by hospitalizations. We included 221 incident JAKi users in the second analysis. We observed 109 ED accesses, 39 hospitalizations, and 64 visits. Injury and poisoning (18.3%) and skin (13.8%) caused ED accesses, and cardiovascular (69.2%) and musculoskeletal (64.1%) caused hospitalizations. The mean costs were 4819 € (607.5; 50,493) per patient, mostly due to JAKi. In conclusion, the JAKi introduction in therapy occurred in compliance with RA guidelines and the increase in costs observed could be due to a possible selective prescription.

## 1. Introduction

This paper is the extended version of the works presented as abstracts to the 37th International Conference on Pharmacoepidemiology and Therapeutic Risk Management (ICPE) Virtual, 23 August 2021 [1,2], and to the 20th International Society of Pharmacovigilance (ISoP) Annual Meeting “Integrated pharmacovigilance for safer patients” 8–10 November 2021 Muscat, Oman (Hybrid meeting) [3]. Rheumatoid arthritis (RA) is an immune-mediated inflammatory disease (IMID) characterized by progressive joint erosion and articular damage. Cytokines play a key role in the pathophysiology of RA, such as interleukin(IL)-1, tumor necrosis factor (TNF), and IL-6 [4]. These cytokines, designated by scientific research as pharmacological targets, have led to therapeutic approaches with different degree of effectiveness. Type I/II receptors are responsible for binding to these cytokines, which carry out their action using the transduction pathway of Janus kinase (JAK) [5]. Therefore, the JAK represents an important pharmacological target—to achieve control of the pathologic response characterized by immune-based inflammation [6]. Two generations of JAK inhibitors (JAKi) were developed over time [6]. The first one includes drugs featuring by a non-selective inhibition of JAK subtypes [6,7], while the second involves drugs characterized by a selective inhibition of JAKs [6,8]. In Italy, the JAKi approved for the treatment of RA, classified as targeted synthetic disease-modifying antirheumatic drugs (tsDMARDs), were baricitinib and tofacitinib as regards the first generation, and filgotinib and upadacitinib for the second generation [9,10]. In this study, we focused on the first generation of JAKi approved for rheumatoid arthritis.

In Italy, tofacitinib has been available since March 2017 [9], as tablets for oral administration in a dose of 5 mg/twice a day [11]. Data on its use are monitored continuously. In February 2021, the public access repository of adverse drug reactions (ADRs) of the Italian Medicines Agency (AIFA, report Reazioni Avverse dei Medicinali, RAM system) reported 171 suspected ADRs for tofacitinib [12]. The most reported ADRs has been observed for the System Organ Class (SOC) general disorders, followed by gastrointestinal complications and infections. On January 2020, the Pharmacovigilance Risk Assessment Committee (PRAC) of the European Medicines Agency (EMA) recommended using tofacitinib in patients older than 65 years old only when no alternative treatment was available due to the increased risk of serious infections, major adverse cardiovascular events, and malignancies [13,14,15]. Indeed, the post-marketing surveillance revealed these signals, particularly with the dosage of 10 mg bis per day [6].

Baricitinib was authorized in Italy in February 2017 [10] as 2 mg and 4 mg film-coated tablets [16]. Serious infections leading to hospitalization or death, including tuberculosis and bacterial, invasive fungal, viral, and other opportunistic infections, were recorded in clinical trials, in addition to high levels of low-density lipoproteins (LDL) cholesterol (34%), upper respiratory infections (15%), and nausea (3%) [16]. In the AIFA—RAM system, 443 ADRs were recorded to be associated with baricitinib, of which the infections were predominates [12]. However, due to its subsequent availability on the market as compared with tofacitinib, baricitinib was initially less used but showed a safety profile in line with the ADRs listed on label [6].

The clinical recommendations for the treatment of RA depict the tsDMARDs as a second-line therapy following a therapeutic failure of the monotherapy with conventional synthetic (cs) DMARDs [17,18]. On the contrary, at the time of the first authorization of tsDMARD on the market, the RA clinical guidelines recommended their use after at least one biologic (b) DMARD [19]. In particular, rheumatologists have to check periodically whether progresses have been achieved in treated patients. The monitoring has to be performed quarterly for the evaluation of a reduction in disease activity until obtaining an in-target disease within 6 months. When a therapeutic strategy fails in the disease outcome, the swap to another drug class of DMARD is recommended, for instance, from a bDMARD to a tsDMARD [6,19].

A recent review on the RA pharmacotherapy showed how the research is working to move from an incurable to curable disease and in particular, this revealed that Italy is among the ten countries with the most prolific publications on the field. As regards the JAKi, the authors highlighted that despite their good safety profile, gastrointestinal, pulmonary, hematological, hepatic, and infective ADRs associated with a tsDMARD have been reported when JAKi was administered after a bDMARD [20].

This study (LEONARDO study) aimed at characterizing the incident RA users of JAKi in the immediate post-approval period (2018–2019). We described drug-utilization and accesses to the healthcare system services before and after the initiation of JAKi, also displaying the direct costs associated with these resources according to the perspective of the regional healthcare system (RHS).

## 2. Results

### 2.1. First Analysis

We identified 450 incident JAKi users in the inclusion period, and according to the exclusion criteria, the final cohort included 363 patients (Appendix A). The mean age of patients was 61.5 (standard deviation, SD = 13.6) years old, and females were 80.7%. We found 21.5% and 78.5% of incident tofacitinib and baricitinib users, respectively. In Table 1, the baseline characteristics and distribution of JAKi users are displayed.

When the history of DMARD use was explored, we observed 8% of incident JAKi users without previous prescription of any DMARD, and 79% with a record of csDMARD supply. The most frequent csDMARD dispensed were hydroxychloroquine (44%), methotrexate (42%), and leflunomide (33%). Among bDMARD users, patients with a history of anti-TNF were 60%, with most receiving etanercept and adalimumab, while patients with bDMARDs with an alternative mechanism of action (MOA) from TNF inhibitors were 41.9%, with most using abatacept and tocilizumab (Appendix A). About 30% of patients used the first JAKi as a second-line treatment (Table 2). The distribution of JAKi dispensations stratified by DMARD history is shown in Appendix A.

The mean time from the first ever DMARD and the first ever bDMARD to the JAKi use was 7.2 (SD 3.3) and 4.5 (SD 3.2) years, respectively. Appendix A shows details about the history of DMARD use according to years preceding the JAKi use.

Table 3 displays the mean number of events per patient/year as regard emergency department (ED) accesses, hospitalizations, and RA visits, ranging between 0.45 and 0.62, 0.23 and 0.31, and 1.02 and 1.44, respectively.

The overall direct healthcare costs from the RHS point of view varied from 1,551,981 € to 1,898,227 €. When the single item of costs was analyzed, we found that costs of hospitalizations ranged from 271,317 € in the 5th year before the cohort entry to 521,431 € in the 1st year before the cohort entry. These corresponded to a mean cost of hospitalization per patient/year of 747.4 € in the fifth year before the cohort entry and 1436.5 € in the first one (Table 4).

### 2.2. Second Analysis

We identified 276 incident JAKi users and we included 220 patients in the final cohort (Appendix A). In Table 5, the distribution and baseline characteristics of the JAKi users are shown.

As regards the healthcare service accesses in the first 6 months of JAKi utilization, we observed 109 ED admissions, 39 hospitalizations, and 64 RA visits. All baricitinib users have a RA visit (n = 38), while there were none for tofacitinib users (Appendix A and Table 6). 

Among 54 patients with at least one ED admission, 14 were males, 46 used baricitinib, and the mean time to the first ED admission was 73.5 days (SD 54.1) in the overall cohort. When hospitalizations were observed, we found that out of 28 patients with at least one hospitalization, nine were males, and six used tofacitinib. In the overall JAKi users, we observed 89.6 (SD 54.8) days to the first hospitalization.

The most frequent causes of ED access associated with JAKi use were injury and poisoning (20 cases; 18.3%), diseases of the skin and subcutaneous tissue (15 cases; 13.8%), and circulatory disorders (12 cases; 11.0%) (Appendix A). The reported hospitalizations were diseases of the circulatory system (27 cases; 69.2%), musculoskeletal and connective tissue disorders (25 cases; 64.1%), and disease of the skin and subcutaneous tissue (15 cases; 38.5%) (Appendix A).

Total direct costs were 1,054,530 €, including 887,946 € of drugs, 165,624 € of hospitalizations, and 960 € of RA visits. The corresponding mean cost per patient was 4793.3 € (607.5; 50,306), involving 4036.1 € (607.5; 8387.9) for drugs, 752.8 (0; 43,811) € for hospitalizations, and 4.4 (0; 60) € for RA visits (Appendix A).

## 3. Discussion

This descriptive exploration of baricitinib and tofacitinib utilization showed that these new drugs were used in accordance with labels [11,16] and clinical guideline recommendations [17]. In line with these, JAKi was used at least as a second-line treatment pharmacological approach when csDMARDs alone failed in controlling RA disease. In this study, overall, 334 patients (92.0%) had a history of previous use of DMARDs and the JAKi use occurred as the second, third, or fourth line of treatment. Only 29 patients (9.0%) had JAKi as the first prescription. We considered that JAKi could have been used in these patients as a first therapeutic choice, but at the same time, we hypothesized that these are patients who have acquired csDMARDs in a private regime or in other regions.

In our study, females represent the majority the JAKi users, and this distribution reflects the women/men ratio reported in the medical literature for RA, about 3:1 [21]. In addition, we observed a higher number of baricitinib users than tofacitinib ones. This is explained by the different years of approval of these drugs: baricitinib obtained approval earlier than tofacitinib for co-payment purposes in the Italian healthcare system [22,23].

In the first analysis, the overall costs slightly increase due to the costs associated with hospitalizations that almost doubled over the years, while the other resources show similar costs over the years, as well as for the mean direct cost per patient/year that increases from the fifth to the second year preceding the cohort entry, in this case owing to the cost of hospitalizations.

Taking into account the healthcare facilities used from the 5th to the 2nd year before the first JAKi dispensation, we observed an increasing trend over the years, but in the year before the JAKi introduction, the ED accesses and the RA visits decreased with the exception of the hospitalizations that remained constant. We can therefore speculate that the JAKi were prescribed preferentially to patients who had better clinical conditions in the year preceding the cohort entry, hypothesizing a selective prescription of JAKi in the first period of their approval. This phenomenon is known as selective prescription [24,25,26,27], and it can occur when clinicians prescribe drugs of new market introduction in two different situations. In the first, they prescribe to patients with a low burden of diseases, and in the second, they prescribe to those with no response to previous treatments. An example of the first option in RA was reported by a study conducted by Frisell T. et al., 2017 [24]. In this study, the authors evaluated bDMARDs utilization and compared anti-TNF with MOA users to assess whether patient characteristics could drive the choice of bDMARD in clinical practice. They highlighted that the anti-TNFs were prescribed to patients who were in better clinical conditions. Indeed, rituximab (a MOA drug) initiators were more often seropositive, had a long illness history, and had a slightly higher erythrocyte sedimentation rate (ESR) than anti-TNF users. Tocilizumab (another MOA bDMARD) initiators had more active disease with higher ESR and C-reactive protein (CRP) than anti-TNF patients did. Finally, compared with those who started anti-TNF, rituximab and abatacept (MOA drugs) initiators depleted more healthcare resources before treatment started [24]. As regards the second option, to the best of our knowledge no evidence was found in RA, but the selective prescription can be observed in the first years of any drug approved, and as for JAKi, another example can be seen in diabetes. Compared to the traditional oral antidiabetic drugs, sitagliptin was prescribed to older patients with comorbidities and no responders to previous treatments [25,26,27].

When the total direct costs from the fifth to the second year before the first JAKi prescription were assessed, we observed an increasing trend over the years. As regards the hospitalizations, we found an increase in the related costs with a constant number of these events in the year before the cohort entry. This could be explained by the occurrence of serious events requiring hospitalization that had affected these costs [28]. Noteworthily, since drug and visit costs decreased in the year before the cohort entry, this observation can confirm the hypothesis of a selective prescription. Harnett J. and colleagues in the United Stated also performed a cost assessment in the 12 months before and after tofacitinib initiation in RA patients. They found that the trend of mean medical costs decreased from the pre to post-index period [29]. However, the population, the observation period, and the different healthcare systems make these data highly different from ours. Another study, evaluating healthcare resources in a cohort of Colombian RA patients exposed to bDMARDs or tofacitinib and using claims data and electronic medical records, showed that the majority (97.2%) of the direct costs were related to drugs [30], as occurred in our study, highlighting that the cost of drugs involved about 84% of the overall direct costs.

In the results of the second analysis measuring the first 6 months of JAKi use, we observed that baricitinib was the most frequent JAKi prescribed as compared tofacitinib, and that women were the gender most frequently reported. These characteristics were also found in the first analysis and were in line with the literature [21].

As regards the utilization of RHS services, we found differences between genders in the time to ED access and hospitalization, being longer in females than males, and in the distribution of hospital admissions (both ED and hospitalizations) being more frequent in males than females. When the type of JAKi was considered, a similar distribution in hospitalization was observed both for baricitinib and tofacitinib, while baricitinib users had a shorter time to first hospitalization than tofacitinib ones.

The most frequent causes of ED accesses and hospitalizations were in line with the natural history of the disease or with the safety profile of the JAKi class. Among the most relevant events, cardiovascular ones deserve discussion. Indeed, although cardiovascular problems are known to represent a complication of RA, adverse events such as venous thromboembolism and deep vein thrombosis have also been reported for tofacitinib [17,31]. On February 2019, a Drug Safety Communication of the Food and Drug Administration (FDA) confirmed the presence of the cardiovascular risk associated with the intake of tofacitinib to a dosage of 10 mg in RA patients, as shown by a post-marketing safety clinical trial (A3921133) [12]. This study, assessing the safety profile including cardiovascular, oncological, and infectious impairments associated with the 5 mg and 10 mg BID of tofacitinib as compared with the anti-TNF drugs, highlighted an increased risk of blood clots in the lungs and death in RA patients with the higher dosage [32]. The revision of this study by the EMA, which also included results from earlier studies and consultations with experts, led to the recommendation of using tofacitinib with caution in patients that were high risk for events such as blood clots [13,15]. Preliminary results from a safety clinical trial demonstrated an increased risk of serious heart-related diseases and cancer with tofacitinib compared to TNF inhibitors [33]. However, these alleged side effects, which have had to be evaluated in long-term studies, overlap with RA cardiovascular complications, representing one of the most frequently reported complications [34]. Indeed, in the RA population, cardiovascular events are 1.5–2 times more frequent than those in the general population [35]. Therefore, it is difficult to make a distinction between JAKi side effects and RA complications, especially because in most patients JAKi is used as a second, third, or fourth-line treatment when the disease history is advanced, and the patients are already affected by comorbidities. However, the label of tofacitinib was updated for the risk of major cardiovascular problems and cancer in light of the results of this safety review performed by EMA [14,15].

The costs obtained in the second analysis, both total direct and stratified for a single item, are not comparable with those observed in the first one due to differences in the two populations. In the first 6 months of the use of JAKi, we found that few patients (about 20%) showed RHS facility use. Since out of 220 patients, only 28 had at least one hospitalization and 54 had an ED admission, the use of JAKi did not have a significant impact on the disease burden as regards the safety profile of patients. This could be due to a good response to therapy or to our hypothesis of the occurrence of the selective prescription of JAKi. Nevertheless, further assessments will be needed to confirm our findings by designating ad hoc studies when a longer follow-up and population will be available and the prescriptive habit will be strengthened.

This study has some limitations. First, the use of healthcare administrative data makes it difficult to assess the causality of the observed causes of RHS admissions, as well as an association with drugs. Second, the causes of hospitalizations and ED admissions were not stratified for single JAKi, and therefore no safety conclusions could be pointed out for tofacitinib or baricitinib separately. Third, the ethnicity of patients was not evaluated, since information was not collected in these kinds of databases. Finally, as a point of strength, this is a descriptive study of what really happened soon after the approval of JAKi in Tuscan clinical practice. It is thus a picture of their use, and the related RHS facilities were provided.

## 4. Materials and Methods

### 4.1. Study Design and Data Source

The LEONARDO study, EUPAS35746 [36], is a descriptive, population-based retrospective cohort study. We retrieved data recorded in the administrative healthcare databases of Tuscany. The population of Tuscany, with about 3 million inhabitants, benefits from the universal regional public health system, with a single payer, for which the facilities provided to patients at the regional level are collected in a uniform process from several clinical centers that provide facilities. Each center collects data according to a single regional protocol throughout the territory. For this purpose, we linked four different repositories: hospital discharge records (reporting the cause of hospitalizations coded by the International Classification of Diseases, 9th revision (ICD-9 codes), hospital admission date, discharge date, hospital stay costs); ED accesses (including diagnoses, ICD-9 codes, causing ED admission, ED admission date, and discharge date); drug supplies (Anatomical Therapeutic Chemical Classification System (ATC codes), drug supply date, doses, drug costs); specialist encounters (RA visits, date, and cost of visits). Information was linked and analyzed through an anonymous unique patient code.

### 4.2. Study Population

We identified two different study cohorts. We selected two different time intervals for the two analyses because, in the first analysis, we characterized patients based on drug and healthcare utilization history, while in the second analysis, we assessed the outcomes in the follow-up.

In the first analysis, the cohort entry was defined by the first supply of a JAKi recorded from 1 January 2018 to 31 December 2019.

In the second analysis, the first dispensation of a JAKi was selected from 1 January 2018 to 31 December 2019. We included patients only when at least six months of follow-up was available. The date of the first JAKi supply identified the index date.

In both groups, we excluded patients with less than 10 years of history data preceding the index date (look-back period), diagnosis of cancer, or anti-neoplastic drugs used in the look-back period, and young (≤18 years old) patients at the index date. Furthermore, patient observation was censored at the end of the follow-up or death, whichever came first.

### 4.3. Data Analysis

In the first analysis, we described the distribution of the incident JAKi users in the study period and the DMARD supply history. We investigated the drug class (csDMARD and bDMARD) and the single drug (Appendix A). We estimated the mean time from the first-ever csDMARD and the first-ever bDMARD supply to the first JAKi dispensation. Direct healthcare costs for JAKi users over the five-years preceding the cohort entry were calculated. Overall direct healthcare costs and costs associated with the different healthcare services (dispensed DMARDs, hospitalizations, and RA visits) were evaluated.

In the second analysis, the distribution of ED admissions, hospitalizations, and RA visits (both overall and stratified by drug) were described in the follow-up. We estimated patients with at least one access to ED, hospitalization, and RA visit. In these patients, the mean time to the first ED admission, to the first hospitalization, and to the first RA visit (overall and stratified by type of JAKi and gender) was assessed. Finally, the most frequently reported causes for ED accesses and hospitalizations (the first cause reported in the discharge records) were described. Direct healthcare costs of incident JAKi users in the follow-up were estimated and stratified for dispensed DMARD, hospitalizations, and RA visits.

## 5. Conclusions

In conclusion, our findings showed that in the first period of their market authorization, in Tuscany, the JAKi were used in accordance with the RA clinical guidelines and the utilization of healthcare services was similar between the two drugs. Rheumatologists adopted a selective prescription of JAKi to patients with better clinical conditions in the first period of the availability of these drugs. This study provides a real-world picture of the use of innovative drugs on an Italian population and allows for hypothesizing preferential prescriptive paths that should be studied in future ad hoc studies. The costs of these treatments should be monitored in relation to clinical outcomes in the future to confirm that an optimal cost-effectiveness profile exists.

## Figures and Tables

**Table 1 pharmaceuticals-16-00465-t001:** Incident JAKi users’ characteristics, overall and stratified by the years of the cohort entry (first analysis).

	2019	2018	Overall
JAKi users, n	273	90	363
Tofacitinib, n (%)	77 (28.2)	1 (1.1)	78 (21.5)
Baricitinib, n (%)	196 (71.8)	89 (98.9)	285 (78.5)
Female, n (%)	220 (80.6)	73 (81.1)	293 (80.7)
Age, mean (SD)	61.9 (14.1)	60.3 (11.9)	61.5 (13.6)
Age, median (IQR)	64 (53–73)	60 (52.3–69.8)	63 (52.5–71.5)

JAKi: Janus Kinase inhibitor; n: number; SD: standard deviation; IQR: interquartile rate.

**Table 2 pharmaceuticals-16-00465-t002:** Description of the previous treatment lines of the incident JAKi users (n = 363).

Treatments		JAKi Users, n (%)
*First line*	no history of any DMARD use	29 (8.0)
*Second line*	Overall	113 (31.1)
	csDMARD only	81 (22.3)
	anti-TNF bDMARDs only	25 (6.9)
	MOA bDMARDs only	7 (1.9)
*Third line*	Overall	120 (33.1)
	csDMARDs + anti-TNF bDMARDs	76 (20.9)
	csDMARDs + MOA bDMARDs	29 (8.0)
	anti-TNF bDMARDs + MOA bDMARDs	15 (4.1)
*Fourth line*	csDMARDs + anti-TNF bDMARDs + MOA bDMARDs	101 (27.8)

bDMARDs: biologic DMARDs; csDMARDs: conventional synthetic DMARDs; DMARDs: disease modifying anti-rheumatic drugs; JAKi: Janus Kinase inhibitor; MOA: alternative mechanism of action from TNF inhibitor; TNF: tumor necrosis factor.

**Table 3 pharmaceuticals-16-00465-t003:** Healthcare service facilities accessed in the years preceding the first JAKi dispensation (overall cohort, n = 363).

Patients	ED Accesses	Hospitalizations	RA Visits
*1st year before cohort entry*			
Overall, n	205	113	431
With at least one access, n	122	66	179
Patient/year, mean (min-max)	0.56 (0–7)	0.31 (0–11)	1.19 (0–8)
*2nd year before cohort entry*			
Overall, n	225	111	522
With at least one access, n	117	70	204
Patient/year, mean (min-max)	0.62 (0–6)	0.31 (0–5)	1.44 (0–11)
*3rd year before cohort entry*			
Overall, n	163	98	496
With at least one access, n	108	69	175
Patient/year, mean (min-max)	0.45 (0–6)	0.27 (0–9)	1.37 (0–12)
*4th year before cohort entry*			
Overall, n	166	84	420
With at least one access, n	114	64	153
Patient/year, mean (min-max)	0.46 (0–5)	0.23 (0–3)	1.16 (0–12)
*5th year before cohort entry*			
Overall, n	170	84	370
With at least one access, n	107	67	137
Patient/year, mean (min-max)	0.47 (0–7)	0.23 (0–3)	1.02 (0–12)

JAKi: Janus Kinase inhibitor; ED: Emergency Department; n: number; RA: rheumatoid arthritis.

**Table 4 pharmaceuticals-16-00465-t004:** Direct healthcare costs incurred in the five years before the cohort entry (€).

Direct Costs	Years before Cohort Entry				
	1st	2nd	3rd	4th	5th
Overall (€)	1,676,429	1,898,227	1,754,176	1,621,148	1,551,981
Drugs (€)	1,147,654	1,438,147	1,376,923	1,283,628	1,273,034
Hospitalizations (€)	521,431	450,207	368,797	329,850	271,317
RA visits (€)	7344	9873	8455	7670	7630
Patient/year, mean cost (min–max) (€)	4618.3	5229.3	4832.4	4466.0	4275.4
	(0–96,568.5)	(0–41,415.0)	(0–52,039.5)	(0–30,511.9)	(0–24,265.4)
Drugs (€)	3161.6	3961.8	3793.2	3536.2	3507.0
	(0–15,574.2)	(0–19,892.2)	(0–19,795)	(0–17,745.4)	(0–19,474.4)
Hospitalizations (€)	1436.5	1240.2	1016	908.7	747.4
	(0–93,050)	(0–39,399)	(0–34,677)	(0–26,676)	(0–20,507)
RA visits (€)	20.2	27.2	23.3	21.1	21
	(0–193)	(0–601)	(0–476)	(0–476)	(0–476)

JAKi: Janus Kinase inhibitor; RA: rheumatoid arthritis.

**Table 5 pharmaceuticals-16-00465-t005:** JAKi users’ characteristics: overall and stratified for the years of cohort entry (second analysis).

	2018	2019	Overall
incident JAKi users	89	131	220
Tofacitinib, n	1 (1.1)	41 (31.3)	42 (19.1)
Baricitinib, n	88 (98.9)	90 (68.7)	178 (80.9)
Female, n (%)	73 (82.0)	106 (80.9)	179 (81.4)
Age, mean (SD)	60.1 (11.8)	62.0 (13.7)	61.3 (13.0)
Age, median (IQR)	60.0 (52.0–69.0)	63.0 (54.5–71.0)	62.0 (53.0–71.0)

JAKi: Janus Kinase inhibitor; n: number; SD: standard deviation; IQR: interquartile rate.

**Table 6 pharmaceuticals-16-00465-t006:** Distribution of patients with at least 1 healthcare service admission and time to the first admission in the 6 months of follow-up.

Healthcare Services		Female	Male	Baricitinib	Tofacitinib	Overall
		(n = 179)	(n = 41)	(n = 178)	(n = 42)	(n = 220)
*ED admission*	Patients, n (%)	40 (22)	14 (34)	46 (26)	8 (19)	54 (25)
	Time to first event in days, mean (SD)	77.8 (51.6)	61.2 (61.1)	73.2 (52.9)	75.1 (64.2)	73.5 (54.1)
*Hospitalization*	Patients, n (%)	19 (11)	9 (21)	22 (12)	6 (14)	28 (13)
	Time to first event in days, mean (SD)	92.9 (56.5)	82.6 (53.5)	86.6 (54.7)	100.3 (59.0)	89.6 (54.8)
*RA visits*	Patients, n (%)	30 (17)	8 (19)	38 (21)	0	38 (17)
	Time to first event in days, mean (SD)	63.1 (52.7)	53.0 (47.7)	60.9 (51.3)	NA	60.9 (51.3)

ED: emergency department; n: number; RA: rheumatoid arthritis; SD: standard deviation; NA: not available.

## Data Availability

Data is contained within the article and Appendix A.

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
