# Peer review of "Drug-Utilization, Healthcare Facilities Accesses and Costs of the First Generation of JAK Inhibitors in Rheumatoid Arthritis"

_pharmaceuticals, 2023, doi:10.3390/ph16030465_

Round 1
Reviewer 1 Report
The Manuscript: „ Drug-utilization, healthcare facilities accesses and costs of the first generation of JAK inhibitors in rheumatoid arthritis’’ by Irma Convertino and colleagues retrospectively characterized the new rheumatoid arthritis users of JAKi in the immediate post-approval period. Based on the outcome of the study, the authors described drug-utilization and accesses to the healthcare system services before and after the initiation of JAKi, displaying also the direct costs associated with these resources, according with the perspective of the regional healthcare system. The study is nicely performed with convincing result and supporting discussion. After going through the manuscript, I have a couple of comments for the authors:
1. Being a multi-centre study, how was it assured that the data of patients were uniformly collected/documented? Was a unanimous study protocol developed?
2. Due to the differences between Asian and non-Asian populations in terms of genetic background, the ethnic background of the patients have been known to show varied susceptibility to drugs. Were all patients included in the study of Italian origin or was there a mixed ethnicity of patients? Please mention this point in the manuscript.
Author Response
Reply to Reviewers (1-3)
We are happy to read the positive comments of all three Reviewers. Thanks for appreciating our work. As required, we have introduced in the revised manuscript the revisions suggested by the Reviewers and the Editorial Office.
Replies to Reviewer 1
The Manuscript:”Drug-utilization, healthcare facilities accesses and costs of the first generation of JAK inhibitors in rheumatoid arthritis’’ by Irma Convertino and colleagues retrospectively characterized the new rheumatoid arthritis users of JAKi in the immediate post-approval period. Based on the outcome of the study, the authors described drug-utilization and accesses to the healthcare system services before and after the initiation of JAKi, displaying also the direct costs associated with these resources, according with the perspective of the regional healthcare system. The study is nicely performed with convincing result and supporting discussion. After going through the manuscript, I have a couple of comments for the authors:
Query 1 Being a multi-centre study, how was it assured that the data of patients were uniformly collected/documented? Was a unanimous study protocol developed?
Reply to query 1 This is a population-based study using retrospective data collected in healthcare administrative databases. The regional population of Tuscany, amounting to about 3 million inhabitants, is covered by a universal, single-payer, public health regional system, of which the services dispensed to patients at the regional level were collected uniformly by the different clinical centers providing facilities. Each center collects data according to a regional protocol that is unique in all the territories. We thank the Reviewer for this comment since this gave us the opportunity to implement this point also in the methods section see page 9 lines 295-299 in the revised manuscript attached.
Query 2 Due to the differences between Asian and non-Asian populations in terms of genetic background, the ethnic background of the patients have been known to show varied susceptibility to drugs. Were all patients included in the study of Italian origin or was there a mixed ethnicity of patients? Please mention this point in the manuscript.
Reply to query 2 We thank Reviewer 1 for this observation. Ethnicity is not collected in these kinds of databases. Therefore, we could not include this variable in our protocol. We agree with reviewer 1 that ethnicity could be a relevant element in conditioning the effectiveness and safety of drugs. Therefore, we have introduced this limitation in the discussion section. See page 8 lines 286-287 in the revised manuscript attached.

Reviewer 2 Report
The authors compared tofacitinib and baricitinib users, their prescription and healthcare histories, and the costs of treatment. This is a descriptive study based on the information from healthcare databases of Tuscany, Italy.
Comments
1. Abstract: The first sentence of the Abstract should be rephrased.
2. All the abbreviations should be disclosed on first use in the text.
Author Response
Reply to Reviewers (1-3)
We are happy to read the positive comments of all three Reviewers. Thanks for appreciating our work. As required, we have introduced in the revised manuscript the revisions suggested by the Reviewers and the Editorial Office.
Replies to Reviewer 2
The authors compared tofacitinib and baricitinib users, their prescription and healthcare histories, and the costs of treatment. This is a descriptive study based on the information from healthcare databases of Tuscany, Italy.
Query 1 Abstract: The first sentence of the Abstract should be rephrased.
Reply to query 1 We have modified the first sentence of the abstract in accordance, see the revised manuscript in the attachment.
Query 2 All the abbreviations should be disclosed on first use in the text.
Reply to query 2 We have checked all abbreviations as requested, see the revised manuscript attached.

Reviewer 3 Report
The present article evaluates the implications of drug utilization, healthcare facility access, and the costs of tofacitinib and baricitinib in rheumatoid arthritis patients. The topic is relevant but requires modification and detailing of certain aspects to improve the information and structure of the present article.
Shape suggestions
Bibliographic indexes are per se structures of the article and are not linked to any word in the text. Please revise the whole manuscript from this point of view.
Abbreviations are explained when they first appear in the abstract or main text and contribute to making the text easier to read and the information conveyed more efficiently. Once an abbreviation has been established and explained, it will be used throughout the entire manuscript, with the exception of the abstract, where it must be treated separately. Please revise the whole manuscript and explain the abbreviations used directly, without the explanation (e.g., L22- JAKi; L27- DMARD; L48-RA is explained, but then in L56 is used and not abbreviated, etc.)
L50- No need to use capital letters for the listed cytokines.
L104- There is requested a blank space between texts and tables.
The titles of the tables shouldn’t be included in the rows of the table.
L130- There is a need for a blank space after the table.
Remove number 37 from the automated reference list because it does not contain any references.
Content suggestions
Explanations related to the chosen time interval are needed. Why do the two time intervals chosen for the two analyses differ?
L60 – there are several JAKis approved by regulatory institutions that need to be presented, with different actions depending on the types of JAKs.
The aim of the paper must be reshaped in the last paragraph of the introduction, addressing also aspects related to the motivation of choosing the topic of study, the contributions made to the evaluated field, as well as aspects of novelty.
A more detailed presentation of the JAK types correlated with the JAKi's mechanism of action and safety profile is necessary. I suggest checking and referring to: PMID: 36058148 and PMID: 34831081.
Exclusion criteria should also be presented in the main text, not only in the additional material, because their influence on the results obtained is high. Why were there other comorbidities or associated pathologies besides cancer that were eliminated as exclusion criteria?
It is recommended to present certain criteria that, according to international guidelines, make the transition from one class of DMARD to another. Moreover, a brief presentation of current RA therapy is needed.
The conclusions section needs to be improved, as it is deficient in relation to what has been evaluated. Moreover, it is necessary to include future research directions to address the limitations of this study and, moreover, to include estimates on future expenses and measures to reduce costs.
Author Response
Reply to Reviewers (1-3)
We are happy to read the positive comments of all three Reviewers. Thanks for appreciating our work. As required, we have introduced in the revised manuscript the revisions suggested by the Reviewers and the Editorial Office.
Replies to Reviewer 3
The present article evaluates the implications of drug utilization, healthcare facility access, and the costs of tofacitinib and baricitinib in rheumatoid arthritis patients. The topic is relevant but requires modification and detailing of certain aspects to improve the information and structure of the present article.
Query 1 Bibliographic indexes are per se structures of the article and are not linked to any word in the text. Please revise the whole manuscript from this point of view.
Reply to query 1 We have revised all references accordingly, see the revised manuscript attached.
Query 2 Abbreviations are explained when they first appear in the abstract or main text and contribute to making the text easier to read and the information conveyed more efficiently. Once an abbreviation has been established and explained, it will be used throughout the entire manuscript, with the exception of the abstract, where it must be treated separately. Please revise the whole manuscript and explain the abbreviations used directly, without the explanation (e.g., L22- JAKi; L27- DMARD; L48-RA is explained, but then in L56 is used and not abbreviated, etc.)
Reply to query 2 We have introduced explanations for all the abbreviations mentioned.
Query 3 L50- No need to use capital letters for the listed cytokines.
Reply to query 3 We have deleted the capital letters as requested.
Query 4 L104- There is requested a blank space between texts and tables.
Reply to query 4 Text and tables have been separated accordingly.
Query 5 The titles of the tables shouldn’t be included in the rows of the table.
Reply to query 5 We have changed tables as required.
Query 6 L130- There is a need for a blank space after the table.
Reply to query 6 The spaces have been introduced in accordance.
Query 7 Remove number 37 from the automated reference list because it does not contain any references.
Reply to query 7 The typo has been removed.
Query 8 Explanations related to the chosen time interval are needed. Why do the two-time intervals chosen for the two analyses differ?
Reply to query 8 We selected two different time intervals for the two analyses because, in the first analysis, we characterized patients on the basis of drug and healthcare utilization history, i.e. before the first JAKi supply representing the cohort entry; while in the second analysis, we assessed the drug and healthcare utilization in the follow-up, i.e. 6 months after the cohort entry. Since at that time, we have data available until December 31st, 2019, we decided to restrict the inclusion period of the second cohort up to June 30th, 2019 to have at least 6 months of follow-up available where to check for the outcomes of interest. For the first analysis, conducted in the period preceding the first JAKi supply, the lack of a restriction to a minimum follow-up allowed the inclusion of a larger cohort thus making the analysis more robust. We have introduced a brief explanation of the motivation also in the revised manuscript see page 9 lines 308-311 in the attachment.
Query 9 L60 – there are several JAKis approved by regulatory institutions that need to be presented, with different actions depending on the types of JAKs.
Reply to query 9 The focus of the present work is the first generation of JAKi approved for the treatment of rheumatoid arthritis in Italy. Therefore, we have described in the introduction section the two drugs involved in our study, i.e. baricitinib and tofacitinib. Only for completeness of information, we have decided to mention the second generation of JAKi (filgotinib and upadacitinib) but these drugs were not available at the time of the study for rheumatoid arthritis in Italy. According to this, we would prefer not to introduce the description of other JAKi as suggested by Reviewer 3 since this can distract the reader from our focus (i.e. baricitinib and tofacitinib). However, we will comply with the editor's decision, if he/she would request to include such a detailed description in the introduction.
Query 10 The aim of the paper must be reshaped in the last paragraph of the introduction, addressing also aspects related to the motivation of choosing the topic of study, the contributions made to the evaluated field, as well as aspects of novelty.
Reply to query 10 This study arose from the need of the regional health authority managing the healthcare system to verify and quantify the impact of the introduction of these new drugs on the use of regional healthcare resources in terms of health facilities and costs. In our opinion, the elements of novelty and research contribution are more relevant in the conclusions than in the aim section. Therefore, the conclusions have been updated in agreement. (See page 10 lines 344-346 in the revised manuscript)
Query 11 A more detailed presentation of the JAK types correlated with the JAKi's mechanism of action and safety profile is necessary. I suggest checking and referring to: PMID: 36058148 and PMID: 34831081.
Reply to query 11 We thank the Reviewer for this comment since it gave us the opportunity to improve further the JAKi description and the introduction section. We have introduced the required revision using the two references proposed. (See page 2 lines 54-62, lines 73-74, lines 81-83, and lines 94-99 in the revised manuscript attached).
Query 12 Exclusion criteria should also be presented in the main text, not only in the additional material because their influence on the results obtained is high. Why were there other comorbidities or associated pathologies besides cancer that were eliminated as exclusion criteria?
Reply to query 12 The additional material contains only the flowcharts (SM Figures 1 and 2). In the main text, the exclusion criteria are already shown on page 9 lines 317-321 of the materials and methods section. There are no comorbidities other than cancer considered as exclusion criteria. This is because cancer makes subjects particularly frail and difficult to study in a general cohort. Therefore, these subjects are often excluded from real-world cohorts unless this (cancer) is relevant to the specific goals of the study, and this is not the case.
Query 13 It is recommended to present certain criteria that, according to international guidelines, make the transition from one class of DMARD to another. Moreover, a brief presentation of current RA therapy is needed.
Reply to query 13 The information about RA treatment guidelines was already reported shortly in the introduction section of the original manuscript. However, we have added some details as required by the Reviewer (see page 2 lines 88-93 in the revised manuscript attached).
Query 14 The conclusions section needs to be improved, as it is deficient in relation to what has been evaluated. Moreover, it is necessary to include future research directions to address the limitations of this study and, moreover, to include estimates on future expenses and measures to reduce costs.
Reply to query 14 We thank Reviewer 3 for this comment since this allows adding important points to our conclusions. (see page 10 lines 346-348 in the revised manuscript in the attachment).

Round 2
Reviewer 3 Report
The authors have improved their manuscript.